# POCKET-SPECIFIC 3D MOLECULE GENERATION BY FRAGMENT-BASED AUTOREGRESSIVE DIFFUSION MODELS

## ABSTRACT

Autoregressive model is widely adopted to generate 3D molecules which can fit any protein binding pocket. Current autoregressive model suffers from two major drawbacks. First, it is hard to capture local geometric patterns as only one atom is generated at each step. Second, most of the autoregressive models generate atoms and chemical bonds in two separate processes, which causes a number of problems such as incorrect counts of rings, a bias distribution of bond lengths, and inaccurate 3D molecular structures. To tackle this problem, we designed a model, named FragDiff, to generate 3D molecules fragment-by-fragment for pockets. In each generation step, FragDiff places a molecular fragment around the pocket by using E(3)-equivariant diffusion generative models to simultaneously predict the atom types, atom coordinates and the chemical bonds of the fragment. Extensive experimental results confirm our assumption that unifying the atoms and bonds generations could significantly improve the quality of the sampled 3D molecules in terms of more accurate distributions of 2D subgraphs and 3D substructures.

## 1 INTRODUCTION

Drug design has been greatly improved with the assistance of AI (Stokes et al., 2020; Zhavoronkov et al., 2019). Insilico Medicine recently announces the world's first drug designed by AI has entered Phase 1 clinical trial (Zhavoronkov et al., 2019). AI-based drug design has experienced several important stages. The first generation of methods focus on generating molecule graphs by leveraging multiple graph representation techniques (Jin et al., 2018a;b). Later, researchers realized that the biochemical functions of a small molecule is partially determined by its 3D structure so new models are proposed to directly sample molecular drugs in the 3D space (Hoogeboom et al., 2022; Wu et al., 2022). Recently, an increasing amount of generative models have been developed to generate molecules which can bind to the target protein based on the 3D structures of the binding pockets. A straightforward approach is to encode the geometric features of amino acids on the protein pockets and then translate them to a molecule (Skalic et al., 2019; Xu et al., 2021a). The central problem of this end-to-end approach is that it does not explicitly characterize the interactions of atoms between molecules and pockets. Although involving the pockets, the structure complex of the target protein and molecules are missing so it is hard to quantify whether these molecules can dock into the desired pocket. To solve this problem, new models have been proposed to capture the atom-level interactions between molecules and pockets by directly sampling 3D molecules inside the 3D pockets (Masuda et al., 2020; Luo et al., 2021a; Liu et al., 2022; Peng et al., 2022). However, in comparison to the pocket-free generation, pocket-specific models are still at an early stage and suffers from quite a few problems.

Most pocket-specific models rely on the autoregressive process to generate a molecule. The atoms are placed one by one in the pocket and the chemical bonds are predicted by a separate model. This procedure often leads to inaccurate bond predictions and unrealistic 3D structures. For instance, it needs six steps to generate a benzene ring, which is unnecessary and error-prone. A natural solution is to adopt fragment-based generation approach. However, generating fragments is hard because the model has to simultaneously capture the relationship of more atoms and bonds. The diffusion models have achieved the state-of-the-art performance in various tasks (Ho et al., 2020) including 3D molecule generation (Luo et al., 2022; Xu et al., 2021b; Hoogeboom et al., 2022; Wu et al., 2022).

Diffusion models learn the data distributions by randomly adding noise and recovering the denoised data points. Empirically, it is nontrivial to directly apply the diffusion process to the pocket-specific molecule generation. It is hard to keep the local geometric constraints of a sampled molecules (e.g., carbon atoms in a ring on the same plane) as the complex data is still limited.

To address all these problems, we design a new paradigm, named FragDiff, to design pocket-specific 3D molecules by integrating both the autoregressive and diffusion generative processes. One key observation is that a molecule usually contains multiple 3D fragments with higher frequency to appear inside a particular 3D pocket compared to other fragments. Therefore, we adopt the autogressive process to generate molecules at the fragment level instead of the atom level. Fragment is generated based on the diffusion models instead of extracting from manually-curated databases. In this way, the autoregressive process only need to learn to place a relatively small amount of elements and the diffusion models only need to learn how to generate a relative small amount of atoms such that local geometric and chemical constraints are easily captured. It also helps the diffusion model to capture the atom interactions as the interactions within each fragment are much denser and stronger than outside.

## 2 RELATED WORK

**Pocket-specific molecule generation**   Early pocket-specific molecule generation method is to encode the pockets as latent embeddings and translate them into new molecules (Skalic et al., 2019; Xu et al., 2021a). Such methods usually represent the output molecules as 1D strings or 2D graphs, which cannot explicitly capture the interactions between atoms. Advanced approaches focus on simultaneously generating 2D graphs and 3D structures of the molecules given the pockets. Li et al. (2021) designs a ligand network for this task by using docking scores and Monte Carlo Tree search algorithm. The model infers the interaction between pockets and molecules using AutoDock Vina (Eberhardt et al., 2021) rather than from the observed data, and its performance strongly relies on the accuracy of AutoDock. Masuda et al. (2020) utilized 3D Convolutional Neural Network (CNN) to capture the spatial information and leveraged CVAE to sample 3D molecules. The major drawback is the weak expressivenees and the generated molecules are found to possess poor chemical and drug-like properties (Peng et al., 2022). Autoregressive models have been proposed to solve this problem and achieve SOTA performance. Luo et al. (2021a) first propose to learn the probability densities of different atoms in the 3D space inside the pockets and place atoms based on the learned distributions. Liu et al. (2022) construct local frame to place atoms around the pockets. Peng et al. (2022) further utilized E(3)-equivariant GNN and more efficient sampling scheme to predict the atoms and chemical bonds. However, these methods all adopt the atom-by-atom generation strategy, which may lead to unrealistic subgraphs and inaccurate local 3D structures.

**Fragment-based molecule generation**   Molecules usually contain multiple frequent functional groups. There are quite a few works using fragments for pocket-free molecule generation. Jin et al. (2020) design a hierarchical generation models which first generated structural motifs and then decode to atom sets. Podda et al. (2020) utilize GRU to generate SMILES of fragments, and Xie et al. (2021) employ MCMC algorithm to iteratively edit these fragments. Recently, Powers et al. (2022) designs a fragment-based generation model for pocket binding. However, they rely on prior domain knowledge and over-simplified assumptions. Their model can only select from 28 manually-curated fragments which are connected by single bonds. Generating fragments from a pre-defined fragment vocabulary significantly limits the generation abilities. Another limitation is that these methods do not consider the 3D structures of the generated fragments. These two drawbacks motivate us to develop new models which not only generates arbitrary fragments learned from data but also determines the 3D coordinates of the fragments.

**Diffusion model for small molecule**   Although there is no diffusion-based model for pocket-specific molecule generation, it has been already applied to multiple pocket-free molecule design tasks (Xu et al., 2021b; Wu et al., 2022; Hoogeboom et al., 2022; Jing et al., 2022). For instance, GeoDiff applied a model trained based on the diffusion technique to predict the 3D conformations of molecules (Xu et al., 2021b). Hoogeboom et al. (2022) utilizes equivariant diffusion model to generate small 3D molecules, and Wu et al. (2022) integrates physical prior bridges into the diffusion process for molecule generation. In comparison to pocket-free setting, adding the pocket informa-

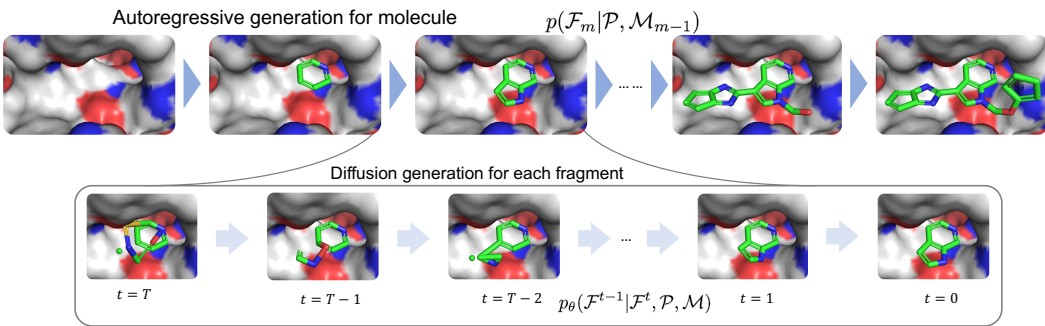

Figure 1: The molecule generation process

tion usually significantly decreases the search space of small molecules and training data. Besides, previous diffusion-based molecule generation models ignored bond generation, which may lead to biased atom connections in the molecules.

## 3 METHODS

### 3.1 DEFINITIONS AND NOTATIONS

A protein pocket is represented as a set of atoms and denoted as $\mathcal{P} = \{(a_i^{(\text{pro})}, r_i^{(\text{pro})})\}_{i=1}^{N^{(\text{pro})}}$ where $a_i^{(\text{pro})}$ and $r_i^{(\text{pro})}$ are the $i$th atom's element type and coordinate, respectively. The molecules and fragments are represented by the atoms and bonds, denoted as $\mathcal{M} = \{(a_i^{(\text{mol})}, r_i^{(\text{mol})}, b_i^{(\text{mol})})\}_{i=1}^{N^{(\text{mol})}}$ and $\mathcal{F} = \{(a_i^{(\text{frag})}, r_i^{(\text{frag})}, b_i^{(\text{frag})})\}_{i=1}^{N^{(\text{frag})}}$, where the symbol $a_i$ and $r_i$ are the $i$th atom type and its coordinate, and $b_i$ is the bonds between the $i$the atom and other atoms. Note that $\{b_i^{(\text{mol})}\}_{i=1}^{N^{(\text{mol})}}$ represents the bonds among the molecular atoms while $\{b_i^{(\text{frag})}\}_{i=1}^{N^{(\text{frag})}}$ represent both the bonds among fragment atoms and the bonds between the fragment atoms and the molecular atoms that the fragment connects to. The pocket-specific molecule generation task is to generate 3D molecules that can bind to any protein pocket. Formally, given a pocket $\mathcal{P}$, the model generate 3D molecules $\mathcal{M}$ from the learned distributions $P_\Theta(\mathcal{M}|\mathcal{P})$ where $\Theta$ is the parameter sets the model has to learn.

### 3.2 MOLECULE GENERATION

#### 3.2.1 GENERATION PROCESS

The molecule generation process is in an autoregressive manner, i.e., by placing fragments around the protein pocket one by one (Fig. 1). In each step, conditioned on pockets and the previously generated molecule, the model generates a fragment by predicting its atom types, atom coordinates, the bonds within the fragment, and the bonds that connect the fragment and the molecules. Formally, the generation process with $M$ steps is denoted as

$$p(\mathcal{M}|\mathcal{P}) = p(\mathcal{F}_1|\mathcal{P})p(\mathcal{F}_2|\mathcal{P}, \mathcal{F}_1)\dots p(\mathcal{F}_M|\mathcal{P}, \cup\mathcal{F}_{<M}) \tag{1}$$

In the $m$th step, the model predicts

$$p(\mathcal{F}_m|\mathcal{P}, \mathcal{M}_{m-1}) \tag{2}$$

where $\mathcal{M}_{m-1}$ is defined as $\mathcal{M}_{m-1} = \cup\mathcal{F}_{<m}$. This probability is predicted by a diffusion probabilistic model and a focal atom predictor. The focal atom predictor determines which atom of the molecule $\mathcal{F}_{<m}$ or protein $\mathcal{P}$ that the new fragment will be generated around, and the diffusion model generated the fragment $\mathcal{F}_m$ around the focal atom.

#### 3.2.2 DIFFUSION FOR FRAGMENT GENERATION

We used diffusion-based generative model to learn the distribution $p(\mathcal{F}|\mathcal{P}, \mathcal{M})$ (Eq. 2, we omit the subscript of autoregressive step in the section). The diffusion probabilistic model defines two

Markov processes, i.e., a forward diffusion process which gradually adds noise to the fragment, and an reverse generative process that denoises the fragment. The fragment at step $t$ of the Markov process is $\mathcal{F}^t = \{a_i^t, \boldsymbol{r}_i^t, \boldsymbol{b}_i^t\}_{i=1}^{N^{(\text{frag})}}$. At $t = 0$, the fragment is the ground-truth, i.e., $\mathcal{F}^0 = \mathcal{F}^{(\text{frag})} = \{a_i^{(\text{frag})}, \boldsymbol{r}_i^{(\text{frag})}, \boldsymbol{b}_i^{(\text{frag})}\}_{i=1}^{N^{(\text{frag})}}$.

**Forward process**   In the forward process, the Markov transition at step $t$ is defined as:

$$
\begin{aligned}
q(\mathcal{F}^t|\mathcal{F}^{t-1}) &= q(\{a_i^t, \boldsymbol{r}_i^t, \boldsymbol{b}_i^t\}_{i=1}^{N^{(\text{frag})}}|\mathcal{F}^{t-1}) \\
&= \prod_i q(a_i^t|\mathcal{F}^{t-1})q(\boldsymbol{r}_i^t|\mathcal{F}^{t-1})q(\boldsymbol{b}_i^t|\mathcal{F}^{t-1}) \\
&= \prod_i q(a_i^t|a_i^{t-1})q(\boldsymbol{r}_i^t|\boldsymbol{r}_i^{t-1})q(\boldsymbol{b}_i^t|\boldsymbol{b}_i^{t-1}) \\
&= \prod_i q(a_i^t|a_i^{t-1})q(\boldsymbol{r}_i^t|\boldsymbol{r}_i^{t-1})[\Pi_j q(b_{ij}^t|b_{ij}^{t-1})].
\end{aligned}
\tag{3}
$$

Here we make an assumption that atom types, bonds and coordinates at step $t$ only depend on their own state at step $t - 1$. More complicated diffusion process could be defined by letting atom types depend bonds and coordinates in the previous step. We leave this generalization in the further work. In the diffusion step, we add Gaussian noise with variance $\beta_{\text{coor}}^t$ to the atom coordinate (Ho et al., 2020) and define the categorical distributions with probabilities of $\beta_{\text{ele}}^t$ and $\beta_{\text{bond}}^t$ to uniformly resample from element types and bond types, respectively (Hoogeboom et al., 2021). If we represent the element types and the bond types as one-hot vectors, the diffusion step can be defined as:

$$
\begin{aligned}
q(a_i^t|a_i^{t-1}) &= \mathcal{C}(a_i^t|(1 - \beta_{\text{ele}}^t)a_i^{t-1} + \beta_{\text{ele}}^t/K_{\text{ele}} \cdot \mathbf{1}) \\
q(\boldsymbol{r}_i^t|\boldsymbol{r}_i^{t-1}) &= \mathcal{N}(\boldsymbol{r}_i^t|\sqrt{1 - \beta_{\text{coor}}^t}\boldsymbol{r}_i^{t-1}, \beta_{\text{coor}}^t\boldsymbol{I}) \\
q(b_{ij}^t|b_{ij}^{t-1}) &= \mathcal{C}(b_{ij}^t|(1 - \beta_{\text{bond}}^t)b_{ij}^{t-1} + \beta_{\text{bond}}^t/K_{\text{bond}} \cdot \mathbf{1})
\end{aligned}
\tag{4}
$$

where $\mathcal{N}$ and $\mathcal{C}$ represent Gaussian distribution and categorical distribution, respectively, $K_{(\text{ele})}$ and $K_{(\text{bond})}$ are the numbers of element types and bond types respectively. $\mathbf{1}$ and $\boldsymbol{I}$ are the all-one vector and identity matrix respectively. Note that the forward process has no learnable parameters. According to Eq. 3 and Eq. 4, as $t \to \infty$, the atom coordinates converge to the standard Gaussian distribution and the element types and bond types converge to the uniform distributions.

**Reverse process**   In the reverse process, the Markov transition at step $t$ is defined as:

$$
\begin{aligned}
p_\theta(\mathcal{F}^{t-1}|\mathcal{F}^t, \mathcal{P}, \mathcal{M}) &= p_\theta(\{a_i^{t-1}, \boldsymbol{r}_i^{t-1}, \boldsymbol{b}_i^{t-1}\}_{i=1}^{N^{(\text{frag})}}|\mathcal{F}^t, \mathcal{P}, \mathcal{M}) \\
&= \prod_i p_{\theta_1}(a_i^{t-1}|\mathcal{F}^t, \mathcal{P}, \mathcal{M})p_{\theta_2}(\boldsymbol{r}_i^{t-1}|\mathcal{F}^t, \mathcal{P}, \mathcal{M})\Pi_j[p_{\theta_3}(b_{ij}^{t-1}|\mathcal{F}^t, \mathcal{P}, \mathcal{M})]
\end{aligned}
\tag{5}
$$

where $p_{\theta_1}(a_i^{t-1}|\mathcal{F}^t, \mathcal{P}, \mathcal{M})$, $p_{\theta_2}(\boldsymbol{r}_i^{t-1}|\mathcal{F}^t, \mathcal{P}, \mathcal{M})$, and $p_{\theta_3}(b_{ij}^{t-1}|\mathcal{F}^t, \mathcal{P}, \mathcal{M})$ are all parameterized through neural networks, whose architecture and training loss will be described in Sec. 3.3 and 3.4. We choose a large $T$ and used the reverse Markov process to generate the fragment as follows,

$$
\begin{aligned}
p_\theta(\mathcal{F}^0|\mathcal{P}, \mathcal{M}) =& p_\theta(\mathcal{F}^0|\mathcal{F}^1, \mathcal{P}, \mathcal{M})p_\theta(\mathcal{F}^1|\mathcal{F}^2, \mathcal{P}, \mathcal{M})\ldots \\
& p_\theta(\mathcal{F}^{T-1}|\mathcal{F}^T, \mathcal{P}, \mathcal{M})p(\mathcal{F}^T|\mathcal{P}, \mathcal{M})
\end{aligned}
\tag{6}
$$

where $p(\mathcal{F}^T|\mathcal{P}, \mathcal{M})$ is defined as a standard Gaussian distribution for fragment atom coordinates and categorical distributions for atom element types and bond types. Technically, we move the focal atom to the origin of the coordinate system at the beginning so that $p(\mathcal{F}^T|\mathcal{P}, \mathcal{M})$ is E(3)-equivariant. We also set a maximal number of atoms of a fragment can contain and define a dummy element type for the fragment atom, so that the model can generate fragments with variant sizes.

## 3.3   MODEL ARCHITECTURE

In this section, we introduce how we define an E(3)-equivariant graph neural network to parameterize the reverse generation model $p_\theta(\mathcal{F}^{t-1}|\mathcal{F}^t, \mathcal{P}, \mathcal{M})$ (Eq. 5) in Sec. 3.3.1 and other auxiliary models in Sec. 3.3.2.

### 3.3.1 Reverse Generation Model

**Inputs** The input of the model is the fragment at step $t$, the molecule and the pocket, i.e., $\{\mathcal{F}^t, \mathcal{P}, \mathcal{M}\}$. The initial features of the protein atom are the element type, the amino acid type it belongs to, and an indicator representing whether it is on the protein backbone. The initial features of atoms are their element types. The atom features of the fragment, the molecule and the pocket are individually embedded to the same dimension using three embedding layers and are composed together as vertex $\boldsymbol{v}_i$. Then we build two sets of edges for the vertices, denoted as $\mathbb{E}$ and $\mathbb{C}$. The set $\mathbb{E}$ contains edges of $k$ nearest neighbors for all vertices, and the edges are initially represented by the edge distances and the types of the two ends, denoted as $\boldsymbol{e}_{ij}$. The set $\mathbb{C}$ contains potential bonds where one ends are fragment atoms and the other ends are the fragment atoms or molecule atoms. The bonds are initially represented by their lengths, atom types of two ends, and bond types (including single bond, double bond, triple bond and none), and passed through an embedding layer to get the hidden vectors denoted as $\boldsymbol{c}_{ij}$. Note that the hidden vectors $\boldsymbol{c}_{ij}$ will also be updated simultaneously with $\boldsymbol{v}_i$ as described in the following, because they will be utilized for bond type prediction.

**Message passing** We use message passing to update the hidden vectors of vertices $\boldsymbol{v}_i$ and bonds $\boldsymbol{c}_{ij}$. Intuitively, the vertex feature is updated using messages calculated from the vertices it connects to, and the bond feature is updated using messages from the other bonds that share the same vertices with it. The updating algorithm is defined as:

$$
\begin{aligned}
\boldsymbol{v}_i &\leftarrow \phi_1(\boldsymbol{v}_i) + \sum_{(i,j)\in\mathbb{E}} \phi_2(\boldsymbol{v}_i, \boldsymbol{e}_{ij}) + \sum_{(i,j)\in\mathbb{C}} \phi_3(\boldsymbol{v}_i, \boldsymbol{c}_{ij}) \\
\boldsymbol{c}_{ij} &\leftarrow \psi_1(\boldsymbol{c}_{ij}, \boldsymbol{v}_i, \boldsymbol{v}_j) + \sum_{(k,i)\in\mathbb{C}} \psi_2(\boldsymbol{c}_{ki}, \boldsymbol{v}_k) + \sum_{(j,k)\in\mathbb{C}} \psi_3(\boldsymbol{c}_{jk}, \boldsymbol{v}_k)
\end{aligned}
\tag{7}
$$

where $\phi$ and $\psi$ represent different fully-connected neural networks.

**Outputs** After $L$ layers of message passing, the model finally used the updated hidden vectors $\boldsymbol{v}_i$ and $\boldsymbol{c}_{ij}$ to predict the denoised fragment, including the categorical distributions of atom element types $\hat{a}_i$ and bond types $\hat{b}_{ij}$ and the mean of the Gaussian distribution of the atom coordinates $\hat{\boldsymbol{r}}_i$ as follows,

$$
\hat{a}_i = \varphi_1(\boldsymbol{v}_i), \qquad \hat{b}_{ij} = \hat{b}_{ji} = \varphi_2(\boldsymbol{c}_{ij} + \boldsymbol{c}_{ji}), \qquad \hat{\boldsymbol{r}}_i = \boldsymbol{r}_i + \Delta\hat{\boldsymbol{r}}_i
$$
$$
\Delta\hat{\boldsymbol{r}}_i = \sum_{(i,j)\in\mathbb{C}} \varphi_3(\boldsymbol{v}_i, \boldsymbol{v}_j, \boldsymbol{c}_{ij})(\boldsymbol{r}_j - \boldsymbol{r}_i) + \sum_{(i,j)\in\mathbb{E}} \varphi_4(\boldsymbol{v}_i, \boldsymbol{v}_j, \boldsymbol{e}_{ij})(\boldsymbol{r}_j - \boldsymbol{r}_i)
\tag{8}
$$

where the symbol $\varphi$ represents neural networks and $\boldsymbol{r}_i$ are the input coordinates of vertex $i$. The outputs of $\varphi_1$ and $\varphi_2$ are the probabilities of element types and bond types, respectively, while the outputs of $\varphi_3$ and $\varphi_4$ are scalars which work like "force" that drag the atom $i$ towards atom $j$ (Luo et al., 2021b; Guan et al., 2021). Such a design also keeps the E(3)-equivariance, which is approved in the Appendix A.3.

### 3.3.2 Auxiliary Models

**Focal predictor** Before the diffusion model generates the fragment, we use a focal predictor to determine around which atom the new fragment should be placed. And all atoms should be translated so that the focal atom is located at the origin before the diffusion begins. The focal predictor is trained together with the diffusion model. After masking, the unmasked atoms that are connected to the fragments are defined as focal atoms. If all molecular atoms are masked, the protein atoms that are the closed to the fragment atoms are defined as focal atoms. The focal predictor uses the message passing defined in Eq. 8 to learn the features of atoms and use two MLPs to classify the focal atoms for protein atoms and molecular atoms.

**Molecule discriminator** To obtain more information about the molecule structures, we build and train a molecule discriminator on a pocket-free 3D molecule dataset to classify whether a molecule is a real one. After the generation of new fragments, the molecule discriminator is used to score the generated molecules, and the molecules with low scores are dropped. Once an unrealistic intermediate molecule occur, it will be filtered out immediately. The molecule discriminator is trained in

the contrastive manner: for a real molecule, we create a fake counterpart by adding noise. Then we train the discriminator to discriminate the two molecules. The molecule discriminator uses the same message passing defined in Eq. 8 to learn the features of all atoms and bonds, two max pooling layers to aggregate atom-level and bond-level features, and a final MLP to output a score representing the probability of being a real molecule.

## 3.4 TRAINING

### 3.4.1 TRAINING LOSS

The overall training process is to mask some parts of the molecules and train the model to recover the masked molecules fragment-by-fragment. More specifically, given the structure of a pocket-molecule complex, we first decompose the molecule into molecular fragments (described in Appendix A.1) and then masked a fraction of fragments. The ratio of masked fragments is randomly sampled from a uniform distribution $\mathcal{U}(0, 1)$ and is different for different training epochs. Next we choose the masked fragment that originally connected to the unmasked atoms for the model to recover.

The diffusion model learned the reverse generative process $p_\theta(\mathcal{F}^{t-1}|\mathcal{F}^t, \mathcal{P}, \mathcal{M})$ by approximating the true posterior $q(\mathcal{F}^{t-1}|\mathcal{F}^t, \mathcal{F}^0)$ defined in the forward diffusion process, i.e., by minimizing the KL divergence between the two distributions:

$$\mathcal{L}_{\text{diff}} = \mathbb{E}_q \left[ D_{KL}[q(\mathcal{F}^{t-1}|\mathcal{F}^t, \mathcal{F}^0)\|p_\theta(\mathcal{F}^{t-1}|\mathcal{F}^t, \mathcal{P}, \mathcal{M})] \right] \tag{9}$$

which can be decomposed into three terms:

$$\mathcal{L}_{\text{diff}} = \mathcal{L}_{\text{ele}} + \mathcal{L}_{\text{coor}} + \mathcal{L}_{\text{bond}}$$

$$\mathcal{L}_{\text{ele}} = \mathbb{E}_q \left[ \sum_i D_{KL}[q(a_i^{t-1}|a_i^t, a_i^0)\|p_{\theta_1}(a_i^{t-1}|\mathcal{F}^t, \mathcal{P}, \mathcal{M})] \right]$$

$$\mathcal{L}_{\text{coor}} = \mathbb{E}_q \left[ \sum_i D_{KL}[q(\boldsymbol{r}_i^{t-1}|\boldsymbol{r}_i^t, \boldsymbol{r}_i^0)\|p_{\theta_1}(\boldsymbol{r}_i^{t-1}|\mathcal{F}^t, \mathcal{P}, \mathcal{M})] \right] \tag{10}$$

$$\mathcal{L}_{\text{bond}} = \mathbb{E}_q \left[ \sum_{ij} D_{KL}[q(b_{ij}^{t-1}|b_{ij}^t, b_{ij}^0)\|p_{\theta_1}(b_{ij}^{t-1}|\mathcal{F}^t, \mathcal{P}, \mathcal{M})] \right]$$

For the loss of coordinates, we use the simplified version (Ho et al., 2020), i.e., the MSE loss of the predicted mean of the Gaussian distributions and the true coordinates at step $t - 1$, i.e., $\mathcal{L}_{\text{coor}} = \mathbb{E}_q \left[ \sum_i \|\boldsymbol{r}_i^{t-1} - \hat{\boldsymbol{r}}_i^{t-1}\|^2 \right]$.

The focal predictor uses the binary cross-entropy loss $\mathcal{L}_{\text{foc}}$ and is trained together with the diffusion model. During training, we randomly sample a time step $t$ and optimized the total loss $\mathcal{L}_{\text{diff}} + \mathcal{L}_{\text{foc}}$.

The molecule discriminator is trained using contrastive loss. More specifically, let $\hat{y}_{\text{real}}$ and $\hat{y}_{\text{real}}$ be the predictions for the real and noisy molecules, the loss is defined as $\mathcal{L}_{\text{mol}} = \mathbb{E} \left[ \max(\hat{y}_{\text{fake}} - \hat{y}_{\text{real}} + 0.5, 0) \right]$.

### 3.4.2 PRETRAINING ON THE POCKET-FREE DATASET

The number of pocket-molecule complex pairs in the pocket-specific data is limited compared to numerous 3D molecules (i.e., pocket-free molecule data). To enable the model to learn more information about the distribution of the 2D and 3D structures of molecules, we pretrain FragDiff on the pocket-free dataset and then finetune it on the pocket-specific dataset.

## 4 RESULTS

## 4.1 SETUP

Following previous work (Luo et al., 2021a; Liu et al., 2022; Peng et al., 2022), we use the protein-molecule structure dataset CrossDocked Francoeur et al. (2020) and use the same training and test

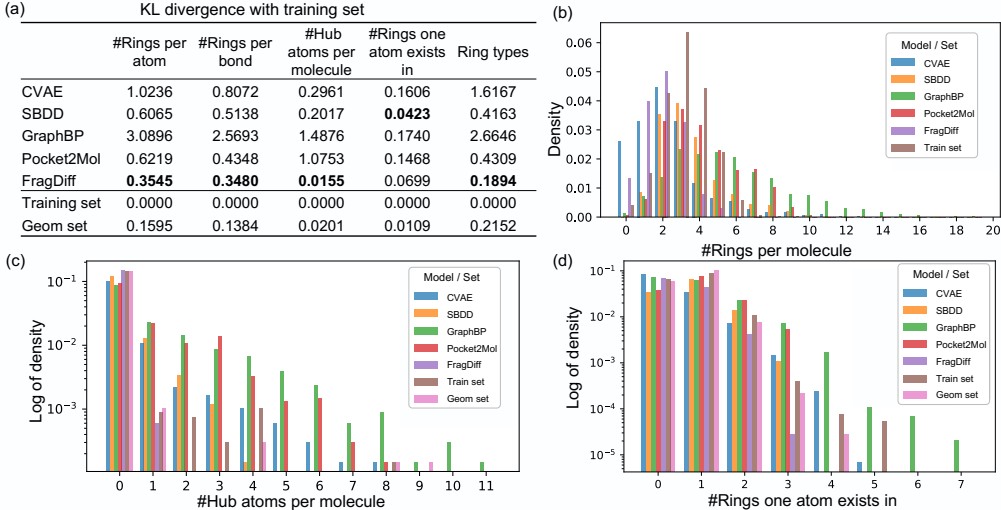

Figure 2: The analysis of distributions of rings. (**a**) The KL divergence of different ring-related metrics between the generated molecules and training set. (**b**) The distributions of counts of rings per molecule. (**c**) The distributions of counts of hub atoms per molecule. (**d**) The distributions of counts of rings that an atom exists in.

splitting. For the pocket-free dataset used for pretraining and molecule discriminator, we use the Geom-Drug dataset (Axelrod & Gomez-Bombarelli, 2022) which contains around 300,000 drug-like 3D molecules. We benchmark FragDiff with CVAE(Masuda et al., 2020), SBDD(Luo et al., 2021a), GraphBP(Liu et al., 2022), and Pocket2Mol(Peng et al., 2022). CVAE uses 3D CNN to encoded the pocket and conditional VAE to generate 3D molecules. SBDD, GraphBP and Pocket2Mol are autoregressive models. The hyper-parameters and training details of FragDiff could be found in Appendix A.2. We choose ten pockets in the test set and sample 100 molecules for each pocket. In the following section, we analyze the 2D and 3D substructures of the generated molecules and compare the distributions with the molecules in the training set. We also conduct the same comparison between the Geom-Drug dataset and the training set for reference.

## 4.2 ANALYSIS OF SUBSTRUCTURES

We show a visualization of the generation process of FragDiff in Fig. 6 in Appendix A.4, where the molecule is generated by sequentially adding fragments. As discussed previously, current models cannot fully capture the local substructures and were inclined to erroneously connect atoms. Since all molecules are connected graphs, incorrect chemical bonds usually lead to the biased ring distributions and form unrealistic rings. Therefore, to evaluate the different approaches, we analyze the rings in the sampled molecules. We count the number of rings in each molecule. Since the number of rings is related to the molecule size, we normalize the number of rings by the numbers of atoms and bonds and compare the distributions of generated molecules with the training set. As shown in Fig. 2(a), the distributions of the number of rings of our method are more consistent with the training set than other baselines. As shown in Fig. 2(b), GraphBP and Pocket2Mol tend to generate excess rings, which is caused by adding bonds in a greedy way to connect atoms.

In addition, we also find such erroneous generation may result in incorrect stack of rings. For instance, too many predicted bonds can form fishing-net like subgraphs. To check this issue, we conduct two experiments: First, if the molecule contains fishing-net like subgraphs, there must be at least one atom existing in three different rings. We define such atoms as "hub atoms" and calculate how many hub atoms each molecule contains. Second, to measure density of fishing-net subgraph, we calculate how many rings an atom exists in, i.e., counts of rings one atom exists in. We showed the distributions of the above two metrics in Fig. 2(c-d) and calculate the KL divergence with the training set (Fig. 2(a)). Overall, FragDiff shows the most similar distributions with the training set, indicating that molecules generated by FragDiff share more realistic substructures than baselines.

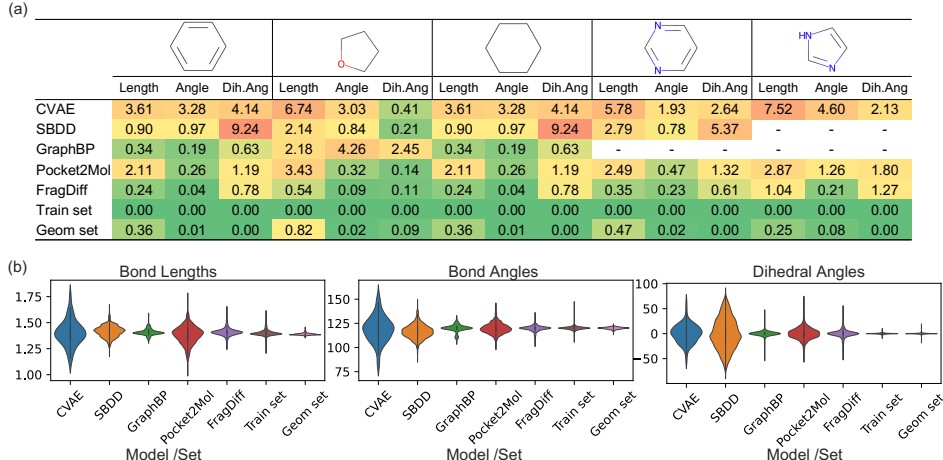

Figure 3: The analysis of 3D distributions of rings. (**a**) The KL divergence of bond lengths, bond angles, and dihedral angles for each ring type. (**b**) The distributions of bond angles, bond angles and dihedral angles of benzene rings.

On the contrary, the baselines GraphBP and Pocket2Mol generate molecules that contain more than four shared atoms, which is rare in both the training set and the Geom set.

We next analyze different types of rings and their 3D structures. We retrieve the top 20 frequent ring types in the training set and then calculated the ratios of the 20 ring types in the generated molecules of different methods (Fig. 7 in Appendix A.4). CVAE and GraphBP generate a large number of rings with only three carbon atoms but extremely small number of benzene rings. We calculate the KL divergence of the distributions of the 20 types of rings to the training set (the last columns in Fig. 2(a)), and find FragDiff has the most similar distributions with the training set. For top ten frequent rings, we analyze their 3D structures by calculating the bond lengths, bond angles and dihedral angles and compare their distributions with training set. We show the KL divergence of their distributions in Fig. 3(a) and Fig. 8 in Appendix A.4 and plot the distributions of benzene rings as an example in Fig. 3. Overall, the distributions of FragDiff are the most similar with the training set, indicating that the molecules generated by FragDiff have more accurate 3D substructures. In the end, we expand the 3D structure analysis to chemical bonds. We analyze the bond lengths of the most frequent bond types (Fig. 9), which also demonstrate that FragDiff generated more accurate bond lengths.

## 4.3 ABLATION STUDY

There are not too many components in our framework. We conduct the ablation study to find whether the pocket-free data contributes to the generation. Apart from the original FragDiff pretrained on the geom dataset, we further construct three ablation models: 1) *No pretraining*: the model directly trained on the pocket-specific dataset; 2) *No pretrain & no discriminator*: the model directly trained on the pocket-specific dataset and does not use the molecule discriminator to filter during sampling; 3) *Guidance*: a model that used different methods to incorporate the pocket-free information. More specifically, apart from a diffusion model directly trained on the pocket-specific dataset, we train an additional diffusion model on the pocket-free dataset. During sampling, the probabilities of the fragments (i.e, the probabilities of the atom element types, atom coordinates and the bond types) predicted by the two diffusion models at each step are weighted averaged, which is inspired by the classifier-free diffusion models (Ho & Salimans, 2022) . The weights for the pocket-specific models and the pocket-free models are set as 0.8 and 0.2, respectively. The sampling process also adopts the molecule classifier to filter molecules.

Our conclusion is that a proper use of pocket-free data can enhance the quality of 3D substructures. Fig. 4 shows four pairwise comparisons where the scatters in the figures represent the KL divergence of the 3D features (i.e., bond lengths, bond angles, and dihedral angles) of different rings and the bond lengths of different bond types. First of all, both pretraining on the pocket-free dataset and

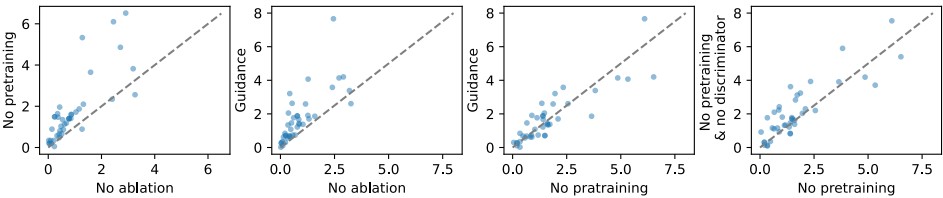

Figure 4: Comparison of KL divergence between different ablation models.

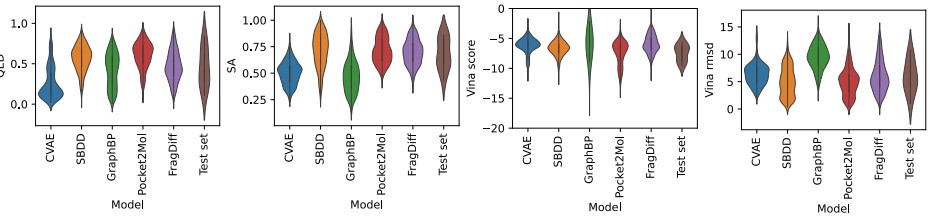

Figure 5: The distributions of QED, SA, Vina score and Vina rmsd of molecules

using molecule discriminator for sampling filtering is beneficial for the 3D structures of the generated molecules, as shown in the first and the last subfigures. The second subfigure also indicates that pretraining the model on the pocket-free dataset is more effective than training a separate diffusion model on the pocket-free dataset. The third subfigure exhibited that the guidance model showed almost the same performance as the model without pretraining, indicating that training a separate diffusion model on the pocket-free dataset might not provide extra information for better molecule generation.

### 4.4 EVALUATION OF OTHER PROPERTIES OF MOLECULES

Following (Peng et al., 2022), we also calculated the QED (quantitative estimation of drug-likeness) and SA (synthetic accessibility score) of the generated molecules and used AutoDock Vina (Eberhardt et al., 2021) to dock the molecules to the pockets. However, we found all these metrics have specific bias. We combined the generated molecules of all these models and those in the test set and found that the SA score of the molecule is correlated the count of atoms and the count of bonds (correlation coefficients are -0.553 and -0.546). The Vina score is highly related to the counts of rings, counts of bonds and counts of atoms (correlation coefficients are 0.713, 0.678 and 0.628). As analyzed above, some baselines tend to generate molecules with excess rings, and thus the Vina scores will be high. Therefore, these metrics are actually not suitable for molecule evaluation in real applications and can only be used as reference. As shown in Fig. 5, we showed the distributions of the these metrics, and found that the distributions of FragDiff was similar to the test set.

## 5 CONCLUSIONS AND DISCUSSION

Generating 3D molecule for protein binding pocket is important for drug design. We designed a fragment-based autoregressive diffusion model for this task. The model generates the molecule by sequentially placing the molecular fragments in the pocket, and for each fragment generation the model leverages diffusion probabilistic model to simultaneously predict the atom types, atom coordinates and bond types. We then detailedly analyzed the 2D and 3D substructures of the generated molecules, and found that FragDiff generated more accurate substructures than baselines.

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

# A APPENDIX

## A.1 FRAGMENTATION OF MOLECULES

We introduce how we break a molecule into fragments.

First of all, we pre-defined a fragment vocabulary set. Note that this fragment vocabulary is only used for fragmentation and training but not for generation. For all molecules in the Chembl dataset Gaulton et al. (2012), we extract all the rings and add them into the fragment vocabulary. Then for each connected graphs of the remaining acyclic molecular graphs, we enumerate all trees given a fragment size and add them into the vocabulary. Finally, we count the frequency of unique fragments in the vocabulary sets and drop out the rare fragments .

After constructing the fragment vocabulary set, we can decompose a molecule into fragment and the maximum fragment size. First, we extract all rings out. Second, for the rings and the remaining connected graphs that are larger than the maximum fragment size, we recursively break the bond so that the two fragments after breaking the bond have the maximum total frequency in the vocabulary. In this way, the molecule is composed into many fragments.

## A.2 HYPERPARAMETERS OF THE MODEL AND TRAINING

In the diffusion model, the number of message passing layers is 6. The hidden dimensions of vertices and bonds are both 256. The hidden dimension of edges is 64. The number of nearest neighbor for edge calculation is 32. For the focal predictor, the difference is that the number of message passing layers is 2 and the hidden dimensions of edges are 32. For the molecule discriminator, the difference from the diffusion model is that the hidden dimension of edge is 128.

For the diffusion parameters, we choose the number of time steps as 3000. The $\beta$ schedulers of all element types, coordinates and bond types are sigmoid Hoogeboom et al. (2021) and the $\beta$s at the start and end steps are $1 \times 10^{-7}$ and $2 \times 10^{-3}$.

We train FragDiff with learning rate $1 \times^{-4}$ using Adam optimizer. The batch size is set as 32. We pretrain FragDiff on the pocket-free dataset for 100,000 iterations and finetune on pocket-specific dataset for 120,000 iterations.

## A.3 E(3)-EQUIVARIANCE OF THE DIFFUSION MODEL

It is straightforward to prove the E(3)-equivariance of the generation model $p_\theta(\mathcal{F}^{t-1}|\mathcal{F}^t, \mathcal{P}, \mathcal{M})$. Assume $T$ is some transformation in E(3) group and are applied to $\{\mathcal{F}^t, \mathcal{P}, \mathcal{M}\}$. The input features and the updated hidden vectors in Eq. 8 remain invariant, thus the predicted element types and bond types are invariant. The predicted coordinates under the transformation are defined as:

$$
\begin{aligned}
\hat{\boldsymbol{r}}'_i &= T[\boldsymbol{r}_i] + \sum_{(i,j)\in\mathbb{C}} \varphi_3(\boldsymbol{v}_i, \boldsymbol{v}_j, \boldsymbol{c}_{ij})(T[\boldsymbol{r}_j] - T[\boldsymbol{r}_i]) + \sum_{(i,j)\in\mathbb{E}} \varphi_4(\boldsymbol{v}_i, \boldsymbol{v}_j, \boldsymbol{e}_{ij})(T[\boldsymbol{r}_j] - T[\boldsymbol{r}_i]) \\
&= T[\boldsymbol{r}_i] + \sum_{(i,j)\in\mathbb{C}} \varphi_3(\boldsymbol{v}_i, \boldsymbol{v}_j, \boldsymbol{c}_{ij})T[\boldsymbol{r}_j - \boldsymbol{r}_i] + \sum_{(i,j)\in\mathbb{E}} \varphi_4(\boldsymbol{v}_i, \boldsymbol{v}_j, \boldsymbol{e}_{ij})T[\boldsymbol{r}_j - \boldsymbol{r}_i] \\
&= T[\boldsymbol{r}_i + \sum_{(i,j)\in\mathbb{C}} \varphi_3(\boldsymbol{v}_i, \boldsymbol{v}_j, \boldsymbol{c}_{ij})(\boldsymbol{r}_j - \boldsymbol{r}_i) + \sum_{(i,j)\in\mathbb{E}} \varphi_4(\boldsymbol{v}_i, \boldsymbol{v}_j, \boldsymbol{e}_{ij})(\boldsymbol{r}_j - \boldsymbol{r}_i)] \\
&= T[\hat{\boldsymbol{r}}_i]
\end{aligned}
\tag{11}
$$

which indicates that $p_\theta(T[\mathcal{F}^{t-1}]|T[\mathcal{F}^t, \mathcal{P}, \mathcal{M}]) = p_\theta(\mathcal{F}^{t-1}|\mathcal{F}^t, \mathcal{P}, \mathcal{M})$, i.e., the outputs are E(3)-equivariant to the inputs.

## A.4 MORE ANALYSIS OF MOLECULES

Figure 6: The generation process of a molecule generated by FragDiff.

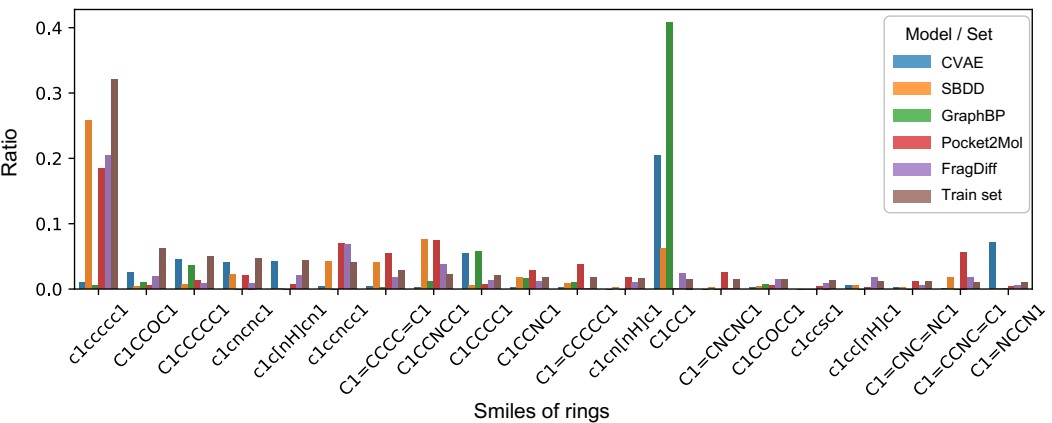

Figure 7: The ratio of the 20 ring types of different models or sets.

| | Length | Angle | Dih.Ang | Length | Angle | Dih.Ang | Length | Angle | Dih.Ang | Length | Angle | Dih.Ang | Length | Angle | Dih.Ang |
|---|---|---|---|---|---|---|---|---|---|---|---|---|---|---|---|
| CVAE | 7.29 | 7.59 | 3.83 | 8.10 | 9.47 | 3.21 | 13.69 | 11.16 | 3.61 | 10.05 | 9.03 | 3.66 | 10.60 | 10.38 | 2.86 |
| SBDD | 2.25 | 1.97 | 6.19 | 1.63 | 2.49 | 4.43 | 4.12 | 2.96 | 1.76 | 7.12 | 2.35 | 0.45 | 2.64 | 0.72 | 0.27 |
| GraphBP | - | - | - | 3.48 | 1.73 | 3.86 | 2.91 | 7.09 | 4.51 | 2.95 | 7.97 | 4.16 | 2.06 | 6.56 | 3.47 |
| Pocket2Mol | 2.47 | 0.53 | 1.14 | 3.35 | 1.13 | 2.23 | 7.11 | 0.70 | 1.11 | 6.36 | 1.19 | 0.35 | 5.52 | 0.45 | 0.53 |
| FragDiff | 0.78 | 0.41 | 0.86 | 1.16 | 0.87 | 1.32 | 1.28 | 0.37 | 0.51 | 2.71 | 0.43 | 0.33 | 3.26 | 0.14 | 0.02 |
| Train set | 0.00 | 0.00 | 0.00 | 0.00 | 0.00 | 0.00 | 0.00 | 0.00 | 0.00 | 0.00 | 0.00 | 0.00 | 0.00 | 0.00 | 0.00 |
| Geom set | 0.19 | 0.00 | 0.00 | - | - | - | 1.05 | 0.05 | 0.03 | 0.52 | 0.03 | 0.22 | 0.99 | 0.02 | 0.06 |

Figure 8: The KL divergence of bond lengths, bond angles, and dihedral angles for more ring types.

(a)

| | CC | CN | CO | C=O | OP | C=C | CF | CS | O=P | C=N | NO |
|---|---|---|---|---|---|---|---|---|---|---|---|
| CVAE | 7.10 | 8.24 | 8.73 | 7.80 | 8.16 | 6.05 | | 10.81 | | 8.92 | 9.88 |
| SBDD | 1.40 | 1.02 | 0.97 | 1.24 | | 0.96 | | | | 4.00 | 3.78 |
| GraphBP | 2.87 | 2.49 | 2.37 | 1.46 | 9.22 | 2.07 | 3.70 | 6.85 | | 3.22 | 4.36 |
| Pocket2Mol | 1.51 | 1.35 | 1.07 | 1.30 | | 2.50 | 3.45 | | | 3.38 | 3.99 |
| FragDiff | 0.44 | 0.41 | 0.68 | 0.57 | 1.59 | 0.48 | 3.19 | 2.40 | 2.45 | 0.85 | 2.91 |
| Train set | 0.00 | 0.00 | 0.00 | 0.00 | 0.00 | 0.00 | 0.00 | 0.00 | 0.00 | 0.00 | 0.00 |
| Geom set | 0.19 | 0.59 | 0.79 | 1.30 | | 0.42 | 0.67 | 0.44 | | 2.16 | 1.88 |

(b)

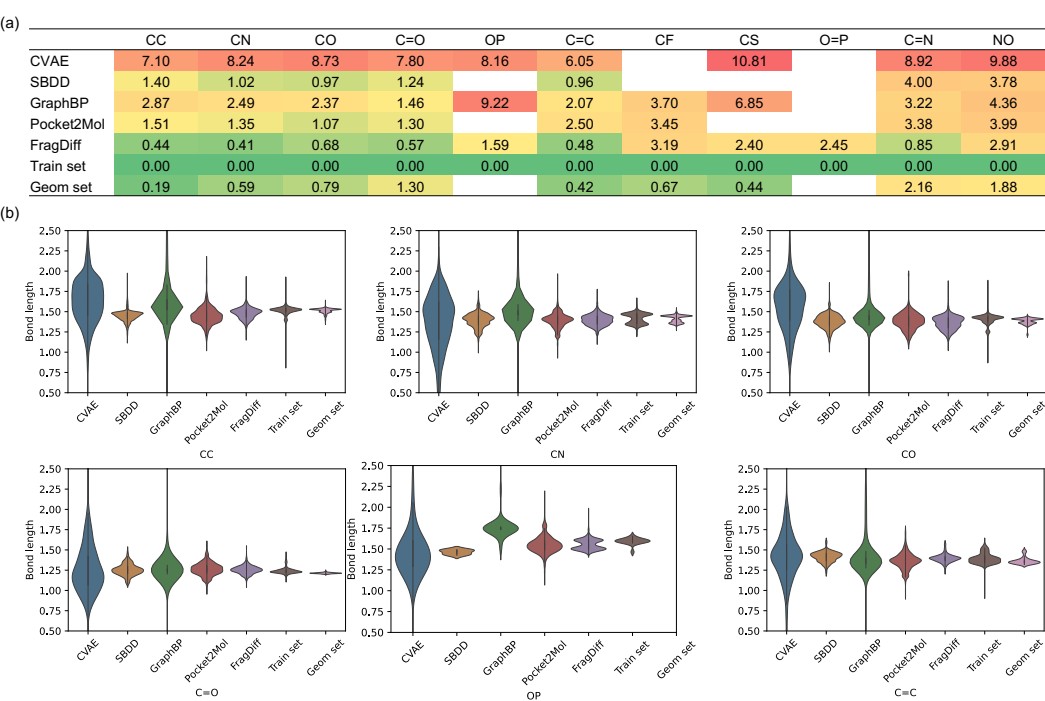

Figure 9: The distributions of bond lengths of different bond types. (**a**) The KL divergence of bond lengths of different bond types with the training set. (**b**) The distributions of bond lengths of the most frequent six bond types.

