# OpenReview forum: "Pocket-specific 3D Molecule Generation by Fragment-based Autoregressive Diffusion Models"
_ICLR.cc/2023/Conference — Submitted to ICLR 2023_

### Official Review · Reviewer_nh1t · 2022-10-17

**Confidence:** 4
**Correctness:** 3
**Technical Novelty And Significance:** 2
**Empirical Novelty And Significance:** 2
**Recommendation:** 3

**Clarity, Quality, Novelty And Reproducibility:**

Clarity: Although the paper is well written, the design of the experiments looks confused, as many ligand-based (unconditional generation) metrics are used, while the proposed model is structure-based generation.

Quality: The paper and the proposed method itself sound good.

Novelty: Incremental, as diffusion models, fragment-based generation, and protein-target based generation are popular recently.

Reproducibility: Most details are provided, but some are not clear, refer to the main review.

**Strength And Weaknesses:**

Strengths:
1. The paper is well-written and easy to follow.

Weaknesses:
1. My most concerning part is the experience section and the performance of the proposed method.
    - There are many metrics for the ligand itself (pocket-free, unconditional), and mostly for the fragments. It is quite confusing as the paper aims to propose a method for protein-specific generation, in which the binding affinity is the most important metric.
    - The selected baselines in the paper are mostly optimized for the binding position/affinity. It is not fair to compare with them in these metrics, especially the proposed method in the paper is optimized for the ring/fragment generation.
    - To compare these ligand-based metrics, it is better to use the ligand-base baseline models.
2. The Vina score from Figure 5 seems poor. I agree that the Vina is not accurate for the binding affinity assessment, but it is a reference. Besides, authors said "the distributions of FragDiff was similar to the test set.", while the distribution of Pocket2Mol is more close to the test set in Vina. And I don't believe close to test-set is better. In short, the current experimental results cannot prove the proposed method is better than previous work in binding affinity assessment.
3. What is "Vina rmsd" ?
4. The inference efficiency. Since the autoregressive model and diffusion model are not efficient during inference, can the authors compare the generation efficiency of the method proposed in this paper with that of other models?
5. Is the proposed a from-scratch protein-target based generation? From Sec. 3.4.1, it seems the training is based on fragment masking, and part of the molecule is retaining. Is part of the molecule is used when performing protein-target based generation?
6. The training is still based on a fragment vocabulary. Did the authors check whether the generative fragments are in the vocabulary or not?
7. The details of "dummy element type" is not provided. How do you define the dummy atoms during training?



**Summary Of The Paper:**

This paper aims to address the problem of generating 3D molecules for a specific protein binding site by integrating both the autoregressive and diffusion generative processes. Specifically, this paper proposes a model FragDiff, which generates 3D molecules fragment-by-fragment auto-regressively. In each time step, the model generates a fragment around the pocket by an E(3)-equivariant diffusion generative model, predicting the atom types, atom coordinates, and the chemical bonds of the fragment.

**Summary Of The Review:**

Overall, this paper looks not ready for now. As a protein-target-based generative model, its experiment did not focus on protein-ligand binding position/affinity, but focused on ligand-based generation (pocket-free). So I recommend rejecting the paper.

---

### Official Review · Reviewer_5r3H · 2022-10-25

**Confidence:** 4
**Correctness:** 3
**Technical Novelty And Significance:** 2
**Empirical Novelty And Significance:** 2
**Recommendation:** 5

**Clarity, Quality, Novelty And Reproducibility:**

**Clarity.** The clarity of the paper could be improved. The model description is rather clear, but there are a few typos across the text, and some figures are illegible. Some architectural choices could be motivated better. Also, I am not fully convinced by the paper motivation as fragment-based models already exist in the literature. Why do we need diffusion models to achieve that?

**Quality.** The conducted experiments show many different aspects of the model, which is compared against a reasonable set of recent generative models. The paper contains a good proportion of both quantitative and qualitative results.

**Novelty.** The use of diffusion models to generate fragments is a new concept that is executed well in the paper.

**Reproducibility.** The implementation is not available. Solely based on the description in the paper, it would be difficult to accurately reimplement the model.

**Strength And Weaknesses:**

Strengths:
- The notation used to explain the model is clear and easy to follow.
- The architectural choices are reasonable, e.g. defining the diffusion of atom types and bond types using a discrete distribution and coordinates with a Gaussian distribution, the use of E(3)-equivariant networks.
- The model equivariance is proven.
- A filtering method based on a discriminator network is prepared and proven to improve the quality of generated compounds.
- The results show that FragDiff can generate realistic 3D structures, at least as well as other recent pocket-based generative models.
- The example of compound generation in Figure 6 shows that the model produces reasonable fragments.

Weaknesses:
- In the abstract, it is argued that current autoregressive models generate one atom at each step. This is certainly not true for all the current models, e.g. see [1, 2]. In the second paper, the fragments are generated using VQ-VAE.
- The statement that “the world’s first drug designed by AI” was recently proposed is exaggerated given the following criticism about the structural similarity to the existing compounds. Also, “drug candidate” would be a better word choice.
- The motivation about using diffusion models to generate fragments is unclear for more. What do we gain exactly?
- For the molecule discriminator, it is said that fake counterparts of molecules are created by adding noise. What kind of noise is added to the molecules? The discriminator training could be described in more detail.
- I am wondering if the filtering procedure based on the discriminator is not too heavy. How many compounds on average are filtered? Could this post-processing method improve the results of other generative models in the benchmark?
- I am confused about the guidance ablation. What was the goal of implementing this ablation study? Were other weights of the pocket-specific and pocket-free models tested as well?
- Some figures are illegible. More specifically, Figure 2 compares the statistics of the generated compounds against the training set, but the columns corresponding to each model are very narrow and difficult to compare with the baseline. It would be far more clear if each model distribution was compared against the training set on a separate plot, or, for the sake of paper length, the baseline distribution could be marked at least in a different way (e.g. a line) to make the other bars wider and easier to distinguish.
- In Figure 2, there are zeros in the row corresponding to the training set. How should I interpret this information, or is it a mistake?

[1] Yang, Soojung, et al. "Hit and lead discovery with explorative RL and fragment-based molecule generation." Advances in Neural Information Processing Systems 34 (2021): 7924-7936.
[2] Chen, Benson, et al. "Fragment-based Sequential Translation for Molecular Optimization." arXiv preprint arXiv:2111.01009 (2021).

Minor points:
- The paper needs proofreading to eliminate grammar errors and typos. Some of the typos: “amount of elements” -> “number of elements”, “around the pocket” -> “inside the pocket”, “the protein atoms that are the closed to the fragment atoms” -> “the protein atoms that are the closest to the fragment atoms”, “bond angles, bond angles and dihedral angles” ->  “bond lengths, bond angles and dihedral angles”


**Summary Of The Paper:**

FragDiff is a new diffusion model that generates compounds for a given binding pocket. The generation is a mix of autoregressive modeling and diffusion that creates molecular fragments. E(3)-equivariant graph neural networks are used to embed molecular graph information. At each diffusion step, the atom types and bond types are represented with discrete distributions and atom coordinates are represented with a continuous distribution. To select the attachment point for the new fragment, a focal predictor is trained. Additionally, a molecule discriminator is trained and used for filtering unrealistic compounds after generation. FragDiff is compared against several recent generative models in terms of structure similarity to the reference set, QED, SA, and Vina scores.

**Summary Of The Review:**

Based on the comments above, I am leaning towards the rejection of the paper.

---

### Official Review · Reviewer_ia3Y · 2022-10-25

**Confidence:** 3
**Correctness:** 3
**Technical Novelty And Significance:** 2
**Empirical Novelty And Significance:** 2
**Recommendation:** 3

**Clarity, Quality, Novelty And Reproducibility:**

Clarity: low, too many equation errors and grammatical errors.

Quality & Novelty: average. The molecule generation process based on the diffusion model and autoregressive model is reasonable.

Reproducibility: difficult. Too many components. Some important parts are missing.

**Strength And Weaknesses:**

Strength:

(1) It is reasonable to leverage the advantages of two kinds of generative models for molecule generation.

(2) The experiment results are impressive.

Weaknesses:

(1) The organization of this paper is not clear. As claimed in the paper, the contribution is the fusion of two kinds of generation models into the same framework. However, the whole technique part, Section 3, is about the construction of diffusion for fragment generation.

(2) How to combine the diffusion model and autoregressive model is not detailed. More content should be added.

(3) Some equations are not correct or confusing.

3.1: Eq.6 is wrong. As the Intermediate variables, $F^{1:T}$, are not integrated out, $F^0$ should be replaced with $F^{0:T}$ in the left part of Eq.6 as well.

3.2: Eq.7 and Eq.8 is wrong. The summation operator over the set of edges, i.e., $\mathbb{E}$ and $\mathbb{C}$, should be normalized.

3.3:  $\psi_1$, $\psi_2$ and $\psi_3$ in Eq.7 and Eq.8 should be different, as they have different types of input.

(4) It seems Eq.9 is the loss for the diffusion model only. Does it mean the autoregressive model is fixed in the whole training process?

(5) Too many grammatical errors, the paper should be double-checked.
Different tenses are mix-used in related work. Singular and plural are mix-used in Section 3.3.2.

**Summary Of The Paper:**

A general framework called FragDiff for pocket-specific 3D molecule generation is introduced. In particular, the generation process is executed in a local-to-global style. Namely, the diffusion model is adopted to generate the local fragment from scratch, while the autoregressive model is used to assemble the fragment into molecules.

**Summary Of The Review:**

Overall, the idea of this paper is average, but it is hard to read due to the wrong equations. Moreover, the organization of this paper is poor. Some important contents are not discussed.

---

### Public Comment · ~Xiufeng_Yang1 · 2023-04-28
**Interesting paper !!  with several questions.**

Hi, nice paper submitted! I went through your paper, from the results, FragDiff could generate better conformational small molecules than all existing methods. Very impressive results, I think.
I have several questions

1. Can Fragdiff generate large complex structures, like proteins?
2. Fragment generation is interesting. But how you controlled the size of generated fragments?
3. You included pocket graph in your model to train FragDiff for generating binding poses.  But in practice, we don't have such large pocket-ligand dataset. Have you considered using FragDiff trained on ligand only to generate binding poses?

Looking forward to your answers. Thank you again for such interesting paper.

---

> ### Author Response · Authors · 2023-05-07
> **Thanks for your interest in our work.**
>
> Thanks for your interest in our work. FragDiff uses the generative diffusion model to generate molecular fragments at each auto-regressive step. Here are the answers to your question:
> 1. It is possible to apply FragDiff for protein generation. The protein is generated in a residue-by-residue manner, and each residue (all atoms of the residue) is generated using a conditional diffusion model. In this setting, the fragments are restricted to the 20 residues and the training data are the protein complex. It might be a good idea to try.
> 2. In each fragment generation step, we set the maximum number of atoms of a fragment (we chose 6 in our paper). We also added a [MASK] type to the atom element type. At the beginning of each diffusion generation process, we randomly sampled 6 atoms from the prior distribution. After the diffusion generation process, the atoms that were decoded as [MASK] type were removed. Therefore,  the diffusion model can generate any fragment with the number of atoms up to 6.
> 3. Actually we did consider the ligand-only dataset. In our paper, we analyzed whether pertaining FragDiff on the ligand-only dataset will have an influence on the final performance.

---

> > ### Public Comment · ~Xiufeng_Yang1 · 2023-05-10
> > **Thanks a lot for your reply.**
> >
> > Do you have any plans to open your source code in the future? I am also looking forward to your updated paper.

---

> > > ### Author Response · Authors · 2023-05-16
> > > **No updated paper**
> > >
> > > Sorry, we do not have an updated version yet and are not going to release the source code of the work. However, you can refer to our new [related work](https://arxiv.org/abs/2305.07508v1) and its source code (to be released). The source code of the new work is adapted from FragDiff.

---

### Decision · Program_Chairs · 2023-01-20

**Decision:**

Reject

**Justification For Why Not Higher Score:**

Numerous questions and clarity issues were brought up by the reviewers; no response from the authors.

**Justification For Why Not Lower Score:**

N/A

**Metareview: Summary, Strengths And Weaknesses:**

The paper introduces a 3d diffusion model for generating fragments that fit to a specific protein pocket.

Unfortunately, despite some promising results and strong leveraging of different generative architectures, all three reviewers had significant concerns with the paper in its current form. This included issues with typos or errors in the text and equations. There are also a number of open questions the reviewers had regarding modeling choices and the experimental setting.

The authors did not choose to respond to the reviewers, so I assume they have decided instead to revise the paper for resubmission elsewhere.